# Megacities as drivers of national outbreaks: The 2017 chikungunya outbreak in Dhaka, Bangladesh

**Ayesha S. Mahmud**[1,9☯]*, **Md. Iqbal Kabir**[2,4☯], **Kenth Engø-Monsen**[3], **Sania Tahmina**[4], **Baizid Khoorshid Riaz**[2], **Md. Akram Hossain**[2], **Fahmida Khanom**[2], **Md. Mujibor Rahman**[5], **Md. Khalilur Rahman**[4], **Mehruba Sharmin**[6], **Dewan Mashrur Hossain**[6], **Shakila Yasmin**[7], **Md. Mokhtar Ahmed**[7], **Mirza Afreen Fatima Lusha**[8], **Caroline O. Buckee**[9]

**1** Department of Demography, University of California, Berkeley, Berkeley, California, United States of America, **2** National Institute of Preventive and Social Medicine, Dhaka, Bangladesh, **3** Telenor Research, Telenor Group, Fornebu, Norway, **4** Directorate General of Health Services, Dhaka, Bangladesh, **5** Dhaka Medical College Hospital, Dhaka, Bangladesh, **6** Ministry of Health and Family Welfare, Dhaka, Bangladesh, **7** Bangladesh Climate Change Trust, Dhaka, Bangladesh, **8** The University of Newcastle, Newcastle, Australia, **9** Harvard T. H. Chan School of Public Health, Boston, Massachusetts, United States of America

☯ These authors contributed equally to this work.
* mahmuda@berkeley.edu

**Data Availability Statement:** Survey data can be requested from the National Institute of Preventive and Social Medicine, Dhaka, Bangladesh (email: info@nipsom.gov.bd); Mobility data can be

## Abstract

### Background

Several large outbreaks of chikungunya have been reported in the Indian Ocean region in the last decade. In 2017, an outbreak occurred in Dhaka, Bangladesh, one of the largest and densest megacities in the world. Population mobility and fluctuations in population density are important drivers of epidemics. Measuring population mobility during outbreaks is challenging but is a particularly important goal in the context of rapidly growing and highly connected cities in low- and middle-income countries, which can act to amplify and spread local epidemics nationally and internationally.

### Methods

We first describe the epidemiology of the 2017 chikungunya outbreak in Dhaka and estimate incidence using a mechanistic model of chikungunya transmission parametrized with epidemiological data from a household survey. We combine the modeled dynamics of chikungunya in Dhaka, with mobility estimates derived from mobile phone data for over 4 million subscribers, to understand the role of population mobility on the spatial spread of chikungunya within and outside Dhaka during the 2017 outbreak.

### Results

We estimate a much higher incidence of chikungunya in Dhaka than suggested by official case counts. Vector abundance, local demographics, and population mobility were associated with spatial heterogeneities in incidence in Dhaka. The peak of the outbreak in Dhaka

requested from Telenor Research, Telenor Group, Fornebu, Norway (email: TelenorResearch@telenor.com). However, due to privacy concerns regarding telecommunications data, permission may be needed from the Government of Bangladesh depending on the proposed usage of the data.

**Funding:** A.S.M. was funded by a Rockefeller Foundation Planetary Health Fellowship during the study period. C.O.B is supported by grants from the National Institute of General Medical Sciences (U54GM088558 and R35GM124715-02). The funders had no role in study design, data collection and analysis, decision to submit the work for publication, or preparation of the manuscript.

**Competing interests:** The authors have declared that no competing interests exist.

coincided with the annual Eid holidays, during which large numbers of people traveled from Dhaka to other parts of the country. We show that travel during Eid likely resulted in the spread of the infection to the rest of the country.

## Conclusions

Our results highlight the impact of large-scale population movements, for example during holidays, on the spread of infectious diseases. These dynamics are difficult to capture using traditional approaches, and we compare our results to a standard diffusion model, to highlight the value of real-time data from mobile phones for outbreak analysis, forecasting, and surveillance.

## Author summary

Chikungunya is an emerging mosquito-borne disease in many parts of the world, with a high morbidity burden. Fluctuations in human density and mobility are important drivers of epidemics, particularly in the context of large cities in low- and middle-income countries, which can act to amplify and spread local epidemics. Here, we first describe the epidemiology of chikungunya in Dhaka, Bangladesh, one of the largest megacities in the world, during an outbreak in 2017. Using data from a household survey, we estimate a much higher attack rate than suggested by official case counts. We then use estimates of population movement from mobile phone data for over 4 million subscribers, to understand the role of human mobility on the spatial spread of chikungunya. We show that regular population fluxes around Dhaka city played a significant role in determining disease risk, and that travel during the Eid holidays likely spread the infection to the rest of the country. Our results show the impact of large-scale population movements, particularly during holidays, on the spread of infectious diseases, and highlight the value of real-time data from mobile phones for outbreak analysis and forecasting.

## Introduction

Human population dynamics underlie the spatial spread of infectious disease outbreaks. Fluctuations in population density and mobility have been shown to be important drivers of the spatial and temporal dynamics for a wide range of pathogens [1–5]. For mosquito-borne diseases, the movement of infected individuals is a critical driver of transmission across spatial scales. While most mosquitoes only travel short distances over their lifespan, particularly the *Aedes* species that transmit emerging viral pathogens including dengue, chikungunya, and Zika, human travel spreads diseases nationally and internationally [6–11]. The movement of people between low and high risk regions has consequences for the spread and maintenance of *Aedes*-borne diseases, and limits prospects for control and elimination [8, 12]. Additionally, for recently emerging infectious diseases such as chikungunya, travel introduces the disease to new locations where the vector exists and local transmission is possible [9, 10, 13].

Understanding the role of population dynamics on infectious diseases is particularly important in the context of rapidly growing urban centers and megacities in low- and middle-income settings, particularly in Asia. Dhaka, the capital of Bangladesh, is one of the densest and fastest growing cities in the world, with more than 18 million inhabitants in the greater

Dhaka area. Similar to other large urban areas in low-income countries, Dhaka serves as a central hub with high connectivity to the rest of the country and internationally. Understanding how infectious diseases spread within and from these urban centers is crucial for the success of epidemic containment measures and for pandemic preparedness. Here, we examine the impact of population dynamics on spatial heterogeneities in chikungunya incidence during a large chikungunya outbreak in Dhaka in 2017, and the spread of the disease from Dhaka to other parts of Bangladesh.

Chikungunya is a viral infection transmitted between people via the bite of infected *Aedes* mosquitoes, and is an emerging disease in many parts of the world [14–16]. While the infection is self-limiting and many individuals are asymptomatic, the clinical symptoms of those who do become ill include fever, rash, and joint pain. These acute symptoms typically last for only a few days, but infection can often result in severe and debilitating joint pain that persists for months [17]. The *Ae. aegypti* and *Ae. albopictus* mosquito vector species that spread the virus are well-established in Bangladesh [18–20], and prior to 2017, three smaller chikungunya outbreaks were reported in various parts of the country, in both urban and rural areas [21–23].

Surveillance for chikungunya in Bangladesh is patchy, with very little information available on the geographic spread of the disease, particularly outside Dhaka. In that respect, Bangladesh is representative of many low-income countries, and innovations that can aid surveillance and intervention planning in the absence of strong reporting systems would be beneficial. The peak of the 2017 chikungunya outbreak in Dhaka coincided with the annual Eid holidays, during which large numbers of people travel from Dhaka to their native region in other parts of the country. While large-scale population movements, including pilgrimages and mass gatherings such as the Hajj, have been implicated with the spatial spread of infectious diseases [5, 24–26], few studies have been able to measure these population dynamics, relying instead on simple diffusion models and simulated mobility data. This is mostly due to difficulties in measuring these large, but short-term, population fluctuations through traditional sources of migration data such as censuses. Recent studies have shown the utility of novel data sources, such as mobile phone call detail records (CDR), in capturing population dynamics in real time at a high spatial and temporal resolution [3, 14, 27]. Mobility estimates from CDR data have been incorporated into epidemiological models to identify seasonal variations in outbreak risk [3, 4], identify areas that may be at high risk for imported cases [8], and highlight the role of mass gatherings on the spatial spread of diseases [5].

Here, we first describe the epidemiology of the 2017 chikungunya outbreak in Dhaka, and examine factors associated with spatial heterogeneities in incidence of symptomatic cases within Dhaka, using epidemiological and larval density data from a household survey. We then estimate the total disease burden, including both symptomatic and asymptomatic cases, using a mechanistic human-mosquito model of chikungunya transmission. We combine the modeled dynamics of chikungunya in Dhaka, with mobility estimates derived from mobile phone data for about 60% of the population in the core of Dhaka city, to understand the spatial spread of chikungunya within and outside Dhaka city. We estimate a much higher incidence of chikungunya in Dhaka during the 2017 outbreak than suggested by official case counts. We show that population flux plays an important role in determining spatial heterogeneities in disease risk within Dhaka city, and that large-scale population movements out of the city during the Eid holidays may have spread the disease widely around the country. We compare our results to the standard diffusion model that is often used in the absence of detailed mobility data, and where possible we compare our predictions to places that did report CHIKV cases during the outbreak, though these were limited. Our results highlight important deviations in actual travel patterns from the standard diffusion model, particularly around holidays, suggesting the utility of these data for disease forecasting.

## Materials and methods

### Ethics statement

All survey respondents provided verbal informed consent. The project was approved by the Institutional Review Board of NIPSOM (Memo number: NIPSOM/IRB/2017/320/1) and the Bangladesh Medical Research Council National Research Ethics Committee (Registration number: 06310082017). The secondary data analysis of the de-identified survey data was determined by the Harvard T H Chan School of Public Health Institutional Review Board (Protocol number: IRB18-1693) as exempt.

### Epidemiological data

A cluster-randomized household survey was conducted in Dhaka city in July 2017 during the peak of the chikungunya outbreak. One hundred clusters were defined in Dhaka city based on administrative boundaries. Approximately 30 households were randomly selected within each cluster, resulting in a total of 3253 households. One individual within each household was asked about symptoms of chikungunya (fever, joint pain, and rash), and the date of onset of the symptoms. Of the suspected cases that met the clinical definition (2518 individuals), a subset were randomly selected for laboratory confirmation of infection with the chikungunya virus (CHIKV). Serum samples were obtained from 1487 individuals and an immunochromatographic test was used to check for the presence of antibodies—immunoglobulin M (IgM) and immunoglobulin G (IgG)—against CHIKV for a random subset of samples (1286 samples). IgM antibodies for CHIKV can be detected 3 to 4 days after clinical onset of illness, and remain detectable for 1 to 3 months post-infection [17, 28]. IgG antibodies, on the other hand, can remain detectable for years [17, 29]. We define an individual to have a confirmed recent infection in 2017, if their sample tests positive for IgM antibodies, as it is unlikely they were infected in a previous outbreak. The delay between infection and the ability to detect IgM antibodies could lead to an under estimation of cases. Due to the retrospective nature of the survey, and the fact that the majority of those reporting symptoms did so more than a week before the survey was conducted, we expect the under-estimation due to the delay to be minimal (and only affecting those who reported symptoms in the final week of the survey). Of the 1286 samples tested for IgM, 895 were positive. Concurrent to the household survey, an entomological team also collected larvae and mosquito samples from pots and containers inside, and in the vicinity of, the house.

### Mobility data

We derived mobility estimates from CDR data from Telenor Group's mobile operator Grameenphone in Bangladesh, with over 64 million subscribers, using methods that have been previously described [3, 4, 8]. The anonymized mobile phone data, covering the the time period between April 1, 2017 and September 30, 2017, was used to estimate movement using the location information encoded in the CDR. Subscribers were assigned to a tower location for each day in the dataset according to their most frequently used routing tower for calls. Trips were calculated based on changes in a subscriber's assigned tower location from the previous day. Due to data regulations and privacy concerns, all data were aggregated temporally (daily) and spatially (to the smallest administrative level, a union) based on tower locations, as described previously [4, 8, 30]. The aggregated data thus consisted of the daily number of subscribers for each union and the total number of trips between all pairs of unions for a 6 month period.

## Modeling drivers of spatial heterogeneity within Dhaka

To understand the epidemiology of the outbreak within Dhaka, we first examined the factors associated with spatial heterogeneities in chikungunya incidence within Dhaka. We aggregated the survey data to 95 administrative units for which we also had information on socioeconomic and environmental indicators (S1 Fig). This spatial aggregation was necessary for correctly matching our data sources, as union boundary definitions have changed several times in Dhaka. To account for the fact that different proportions of suspected cases were sent for lab confirmations in different locations, we estimated the number of chikungunya cases in each location, $O_i$, based on the number of confirmed cases, $C_i$, and suspected cases (reported symptoms but not tested) $S_i$, in each location, and the global ratio of confirmed cases to the number tested, $T_i$:

$$O_i = C_i + S_i \times \frac{\sum_{i=1}^{n} C_i}{\sum_{i=1}^{n} T_i} \tag{1}$$

Here, we make the assumption that the ratio of *actual* cases to suspected cases is homogeneous across space, and that the variation in estimated cases across different locations is driven only by variations in the number of suspected cases, and not by the ratio of confirmed to tested cases. While there may be true variation in $C_i/T_i$ across locations, much of the sample variation in this ratio is likely driven by the small number of tested cases in some locations (S2 Fig). In additional sensitivity analyses, we also estimated $O_i$ by drawing from a binomial distribution with size (number of trials) given by $S_i$ and the probability of a positive case given by the local ratio of confirmed cases to the number tested, $C_i/T_i$. We find that the standard deviation across 100 random draws to be small compared to the overall variation in $O_i$ across locations (S3 Fig).

We used a generalized linear model to assess the impact of demographic, socioeconomic, and environmental factors on estimated cases in each survey location. Specifically, we assumed that our outcome of interest, $O_i$ follows a Poisson distribution with mean, $\eta_i$:

$$O_i \sim Poisson(\eta_i) \tag{2}$$

We modeled the logarithm of $\eta_i$ as a linear combination of a set of independent variables, with the number of households, $households_i$ as an offset. The full model specification is:

$$log(\eta_i) = log(households_i) + \sum_j \rho_j X_{j,i} \tag{3}$$

where $X_{j,i}$ are a set of $j$ independent variables for each location, $i$, and $\rho_j$ are the corresponding estimated coefficients (log of the incident rate ratios). The independent variables in the full model specification are the Breteau index, population density, average income, percentage of area considered to be slum dwelling, percentage of area covered by a water body, the vegetation index, and two measures of mobility and connectedness. The Breteau index (BI) is defined as the number of containers that were positive for mosquito larvae per 100 houses inspected, and was calculated from the entomological survey data. The population size, density, and average income for each location was obtained from *WorldPop* [31]. The percentage of area considered to be slum dwelling was calculated based on the area of each location (from [32]) and the location of slum dwellings (from [33]). Similarly, the percentage of area covered by a water body was calculated based on the location and area of water bodies in Dhaka (from [32]). We

calculated two measures of mobility and connectedness from the CDR data. The *flux* is the average number of people moving in and out of a location daily, normalized by the population size of the location. The network *degree* is the number of edges or connections, for each location in the mobility network, that have weights greater than the average weight of all edges (where weights are defined as the average movement between all pairs of locations, normalized by the number of subscribers in the source location). We use this definition of the network degree since all locations in Dhaka were connected to all other locations daily (i.e. the traditional measure of network degree is the same across all locations). Mobility flux measures relative movements in and out of a location, while network degree is related to how connected a location is to other locations. The maximum likelihood estimates for the coefficients were obtained using the *glm* package in *R*, and robust standard errors were calculated using the *sandwich* package. Incident rate ratios (exponentiated coefficient estimates) and the corresponding 95% confidence intervals (calculated using the delta method) are presented here.

## Human-mosquito transmission model

Since CHIKV is known to cause asymptomatic infections [17], the survey data cannot be used directly to obtain the total burden of chikungunya in Dhaka, since only symptomatic individuals were tested for infection. This is particularly important for estimating importation of cases from Dhaka, as asymptomatic individuals are more likely to travel than symptomatic individuals given the nature and severity of symptoms. We used a mechanistic framework (S4 Fig) to model chikungunya dynamics in Dhaka to estimate daily prevalence of symptomatic and asymptomatic infections. Due to the model complexity, we estimate a single model for all of Dhaka city, rather than a spatially-explicit model.

In this compartmental framework, susceptible humans are exposed to the pathogen through the bite of an infected mosquito. After a latent period, people are either asymptomatically infectious or symptomatically infectious, during which time they can infect susceptible mosquitoes, before recovering and acquiring life-long immunity. In this model, $S$ is the proportion of susceptible humans, $E$ is the proportion of humans in the latent incubation period, $I$ is the proportion that are symptomatically infections, $I^A$ is the proportion that are asymptomatically infectious, and $R$ is the proportion that have recovered. Since infection with CHIKV is thought to confer lifelong immunity [17], recovered individuals remain in the $R$ compartment. For the purpose of fitting to the data, we also tracked the cumulative proportion of symptomatic, $C$, and asymptomatic, $C^A$, infections. In the mosquito population, $S^M$ is the proportion susceptible, $E^M$ is the proportion in the latent phase and, $I^M$ is the proportion infected. The human-mosquito dynamics are modeled as:

$$\frac{dS}{dt} = -\beta_1 S I^M \tag{4}$$

$$\frac{dE}{dt} = \beta_1 S I^M - \lambda_1 E \tag{5}$$

$$\frac{dI}{dt} = \phi \lambda_1 E - \gamma I \tag{6}$$

$$\frac{dI^A}{dt} = (1 - \phi)\lambda_1 E - \gamma I^A \tag{7}$$

$$\frac{dR}{dt} = \gamma(I + I^A) \tag{8}$$

$$\frac{dC}{dt} = \phi\lambda_1 E \tag{9}$$

$$\frac{dC^A}{dt} = (1 - \phi)\lambda_1 E \tag{10}$$

$$\frac{dS^M}{dt} = \mu - \beta_2 S^M(I + I^A) - \mu S^M \tag{11}$$

$$\frac{dE^M}{dt} = \beta_2 S^M(I + I^A) - \lambda_2 E^M - \mu E^M \tag{12}$$

$$\frac{dI^M}{dt} = \lambda_2 E^M - \mu I^M \tag{13}$$

Table 1 describes the model parameters.

The basic reproduction number, $R_0$, is the spectral radius of the next-generation matrix [36]:

$$R_0 = \sqrt{\frac{\beta_1\beta_2\lambda_2}{\mu\gamma(\mu + \lambda_2)}} \tag{14}$$

Note, that the expected number of secondary infections in humans that result from a single infected human, $(R_0^H)$ is $R_0^2$, since two generations are required to transmit an infection from human to human [39]. Since we are primarily interested in human to human infection, we report our estimate of $R_0^H$.

Based on the range of estimates from the literature (see Table 1), we fixed the average latent period of infection in humans, $1/\lambda_1$, at 4 days, average latent period of infection in mosquitoes, $1/\lambda_2$, at 3 days, and the average duration of infectiousness in humans, $1/\gamma$, at 5 days. We assumed a stable mosquito population, such that the mosquito birth and death rates, $\mu$, are the same, and equal to the inverse of the mosquito lifespan, which is fixed at 15 days. We assumed that 80% of the population were susceptible at the start of the outbreak, to account for the fact that Bangladesh had previously experienced chikungunya outbreaks in 2008, 2011 and 2012 [21–23], although these outbreaks were primarily reported in a few small rural communities outside Dhaka. In the survey population, about 5% of individuals had been infected with CHIKV prior to 2017 (IgG positive and IgM negative), although the actual proportion is likely to be higher since only individuals who reported having recent symptoms were tested. Here, we assume that 20% of the population had been infected previously, but varied this proportion in additional sensitivity analyses. The first suspected case in the survey occurred on April 2nd, and the virus was likely circulating prior to that. Thus, the initial incidence in the model was

**Table 1. Parameter definitions for the chikungunya transmission model.**

| Parameter | Definition | Value | Reference |
|---|---|---|---|
| $\beta_1$ | mosquito-to-human transmission rate | estimated | |
| $\phi$ | proportion of infected people who become symptomatic | estimated | |
| $\beta_2$ | human-to-mosquito transmission rate | estimated | |
| $1/\lambda_1$ | average latent period of infection in humans | 4 days | [17, 34–38] |
| $1/\gamma$ | average duration of infectiousness | 4 days | [17, 35, 36, 38] |
| $1/\mu$ | average mosquito lifespan | 15 days | [35, 36] |
| $1/\lambda_2$ | average latent period of infection in mosquitoes | 3 days | [34–36] |

fixed according to the first observation in the survey data. The mosquito-to-human transmission rate, $\beta_1$, the human-to-mosquito transmission rate, $\beta_2$, and the proportion of infected people who become symptomatic, $\phi$, were estimated by fitting the simulated incidence to the survey data.

The weekly incidence in the survey data is given by, $I_w^{obs} = O_w/population$, where $O_w$ was calculated using Eq 1 for each week, $w$, of the outbreak, and *population* is the total number of individuals surveyed, adjusting for the fact that households were not all surveyed simultaneously. To compare the model output to the survey data, we tracked the daily incidence of symptomatic infections ($I_t^{sim} = C_{t+1} - C_t$). We assumed that there is a fixed lag of 2 days for infected individuals to develop symptoms [36]. Thus, the daily incidence of symptomatic infections was assumed to be equivalent to $I_t^{sim}$ from 2 days ago. The daily symptomatic incidence was summed every 7 days to obtain weekly incidence, $I_w^{sim}$. The maximum-likelihood parameter values were estimated by assuming normally-distributed residuals between the observed data and model predictions [36, 38]. We also fit the model assuming a measurement error structure instead i.e. we maximize the likelihood that $I_w^{sim}$ is drawn from a Poisson distribution with mean given by the observed data, and obtained qualitatively similar results (S5 Fig). The log-likelihood was maximized through the Nelder-Mead algorithm within the *optim* package in *R*. We repeated the optimization 100 times with starting values for each parameter drawn from reasonable parameter ranges ($0.2 \leq \beta_1 \leq 0.8; 0.5 \leq \phi \leq 1; 0.2 \leq \beta_2 \leq 0.8$), using latin hypercube sampling to explore the parameter space. The best-fit model was the set of parameter estimates that had the highest log-likelihood value. We show, in additional parameter identifiability analysis with simulated data, that joint estimation of these parameters is possible (S6 Fig) with the observed data.

We calculated uncertainty bounds for parameter estimates by assuming an observational error structure for the incidence timeseries. Specifically, we calculated approximate confidence intervals for the parameter estimates by simulating 200 realizations of the best-fit weekly incidence curve using parametric bootstrap with a Poisson error structure, as has been described in previous studies [40–42]. The best fit values of the weekly incidence, $I_t^{sim}$, was perturbed by adding a simulated error. For each value of $I_t^{sim}$, we simulated $I_t'^{sim}$ which was drawn from a Poisson distribution with mean, $I_t^{sim}$. Parameters were estimated for each of the 200 simulated realizations of weekly incidence. The nominal 95% confidence intervals were constructed from the distribution of the parameter estimates. This approach for constructing the confidence intervals assumes that the only source of noise is observational error.

To test model sensitivity to parameter values, we conducted additional sensitivity analyses using latin hypercube sampling to explore the parameter space and computed partial rank correlations between model parameters and model output. The model output was sensitive to different parameter values (S7A and S7C Fig). However, the partial rank correlation coefficients suggest that the model output variations were largely driven by changes in the values of the parameters we estimated ($\beta_1$, $\beta_2$, and $\phi$), rather than the fixed parameters (S7B and S7D Fig). The maximum-likelihood parameter estimates were stable across a range of initial starting conditions, and the confidence intervals, constructed assuming an observational error structure, were relatively tight. Further, the parameter estimates were stable across a range of values for the initial proportion susceptible (S1 Table), the only fixed parameter for which we had no information from previous studies.

## Quantifying importation of infected travelers from Dhaka

We used the modeled dynamics of chikungunya in Dhaka, and the travel pattern derived from the mobile phone data to estimate the daily number of infected travelers from Dhaka to the

rest of Bangladesh, using methods described in [8]. We first estimated the daily number of infected travelers leaving Dhaka, $T_t$, as $T_t = m_t \times \pi_t$, where $m_t$ is the product of population size and prevalence from the best-fit model (described above) and $\pi_t$ is the daily proportion of people traveling out of Dhaka. To quantify where people are traveling to, we calculated the daily proportion of travelers from Dhaka who travel to each location, $x_{t,j}$.

We quantified importation using mobility estimates from the mobile phone data, as well as a diffusion (gravity) model for comparison. For the model parameterized by the mobile phone data, we estimated $\pi_t$ as the daily proportion of mobile phone subscribers who travel out of Dhaka, and we assume $x_{t,j}$ is equal to the proportion of subscribers leaving Dhaka who go to location, $j$. For the diffusion model, we use a gravity model parameterized with estimates from the literature [43]. We calculated the number of travelers to each location as, $(X_{t,j} = k(p^\alpha \times p_j^\beta)/d_j^\gamma$; where $p$ is the population in Dhaka, $p_j$ is the population in location $j$, $d_j$ is the Euclidean distance from Dhaka to location $j$, and $\alpha$, $\beta$, and $\gamma$ are parameter estimates from [43]. In the diffusion model, $\pi_t$ was given by $\sum_{\forall j} X_{t,j}/p$. The proportion traveling to each location, $x_{t,j}$, was given by $X_{t,j}/\sum_{\forall j} X_{t,j}$.

Since not all importations will lead to local transmission, the number of *effective* infected travelers to each location was sampled from a binomial distribution, similar to a previous study on dengue importation [3], where the probability of "success" (i.e. having an effective infected traveler), $p_{eff}$, was varied from 0.01 to 0.5, and the number of trials was equal to $T_t \times x_{t,j}$. Values of $p_{eff}$ below one reflect the fact that heterogeneities in epidemiological, environmental, and individual factors are likely to result in a smaller number of *effective* infected travelers than $T_t \times x_{t,j}$. In the main results, we assume that on average, 10% of importations result in local transmission, but vary this percentage in sensitivity analyses (S8–S10 Figs).

## Results

### Epidemiology of chikungunya in Dhaka

We estimated a total of 1589 cases of symptomatic chikungunya infection in the 3253 surveyed individuals, based on the proportion of tested cases that were positive for chikungunya and the total number of individuals that reported symptoms (Fig 1A). There was significant spatial heterogeneity in incidence of symptomatic cases within Dhaka. The estimated incidence ranged from 46 to 824 cases per 1000 people across the 95 administrative units surveyed (Fig 1B). Our results indicate that both the abundance of the vector and locals demographics were important drivers of the dynamics of the outbreak. As expected, the Breteau Index, a measure of vector abundance, showed substantial spatial variation (S11 Fig), and had the largest impact—a one unit change in the BI was associated with a 9% [95% CI: 2.5%, 15.8%] increase in symptomatic incidence (Fig 1C). We also found statistically significant positive associations between the incidence rate and population density and connectedness (as measured by the mobility network degree).

We estimated the daily incidence of symptomatic and asymptomatic cases in Dhaka over the course of the outbreak by fitting a mechanistic human-mosquito transmission model to the observed incidence time series. We estimated the following model parameters by fitting the simulated incidence to the survey data: the mosquito-to-human transmission rate, $\beta_1$, the human-to-mosquito transmission rate, $\beta_2$, and the proportion of infected people who become symptomatic, $\phi$. Maximum-likelihood parameter estimates, and associated uncertainties, are shown in Table 2.

The observed survey incidence had two peaks, which may have been driven by the timing of the monsoon rainfall (S12 Fig) [45] or may be due to patchy reporting of cases. Reproducing the bi-modal nature of the outbreak, however, is not possible without increasing model

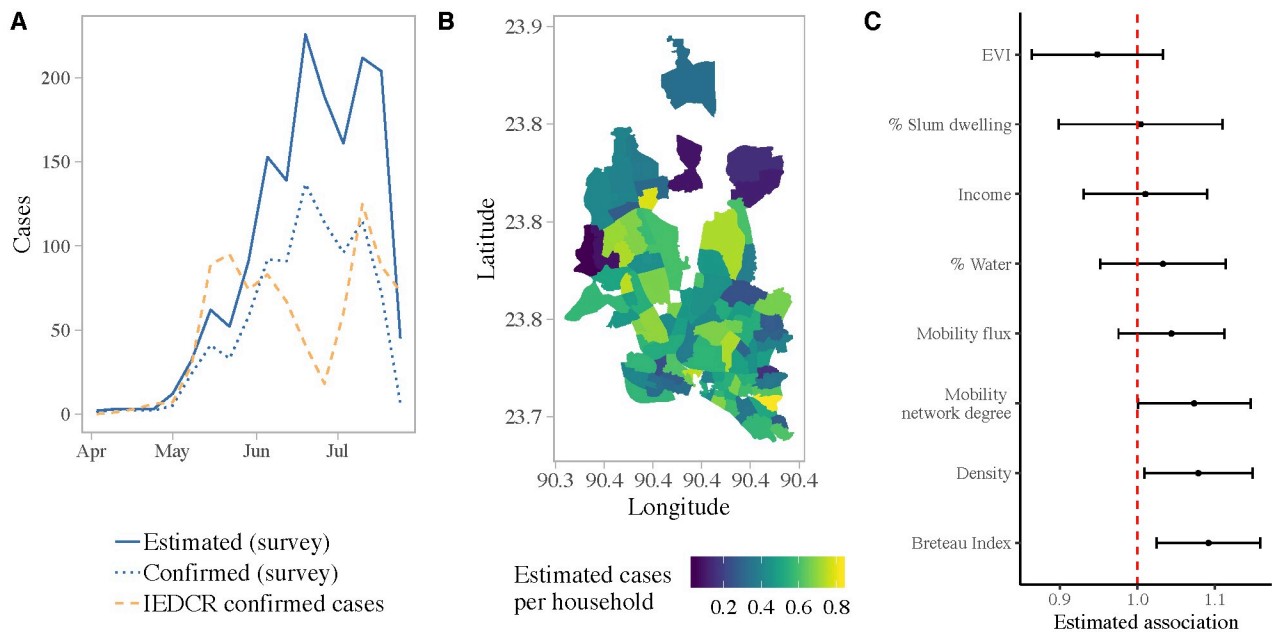

**Fig 1.** (A) Weekly number of confirmed and estimated cases in the survey data. Orange dashed line shows the number of cases reported to the Institute of Epidemiology, Disease Control and Research (IEDCR) over the course of the outbreak. (B) Estimated cases per household for each survey location. (C) Estimated relative risks (incidence rate ratios) from poisson regression model. 95% Confidence intervals were calculated by applying the delta method [44] to robust standard errors.

complexity and adding external forcing in the model. Nonetheless, our best-fit model accurately reproduces the general trend in the observed incidence and the size of the outbreak (Fig 2). The simulated attack rate for symptomatic cases, using the best-fit model parameters, from the start of the outbreak to the end of July, was 51% (standard deviation across simulations with observational error: 1.8), which compares well with the 49% estimated from the survey. According to the best-fit model, 78% [95% CI: 73, 85] of the infections were symptomatic. We estimated a human-mosquito transmission rate of 0.51 [95% CI: 0.32, 0.75]; this is equivalent to the number of mosquito bites per human per day resulting in transmission (accounting for imperfect transmission of the pathogen). Our estimates for both $\beta_1$ and $\beta_2$ are similar to previous estimates for chikungunya using a similar model [36].

Using the best-fit model parameters, we estimated the human-to-human basic reproduction number, $R_0^H$, to be 4.20 [95% CI: 3.83, 4.62], which is similar to previous estimates of $R_0$ for chikungunya [35, 46, 47]. We estimated a peak prevalence of 47 cases per 1000, with the peak incidence occurring in the first week of July. Interestingly, these estimates are far higher than the officially reported number of 13,176 total cases between April and September [45].

**Table 2. Parameter estimates for the best-fit model.** Parameters were estimated by fitting the mechanistic chikungunya model to observed incidence from the survey.

| Parameter | Maximum-likelihood estimate | Approximate 95% CI |
|---|---|---|
| $\beta_1$ | 0.51 | (0.32, 0.75) |
| $\phi$ | 0.78 | (0.73,0.85) |
| $\beta_2$ | 0.16 | (0.10,0.29) |
| $R_0^H$ | 4.20 | (3.83, 4.62) |

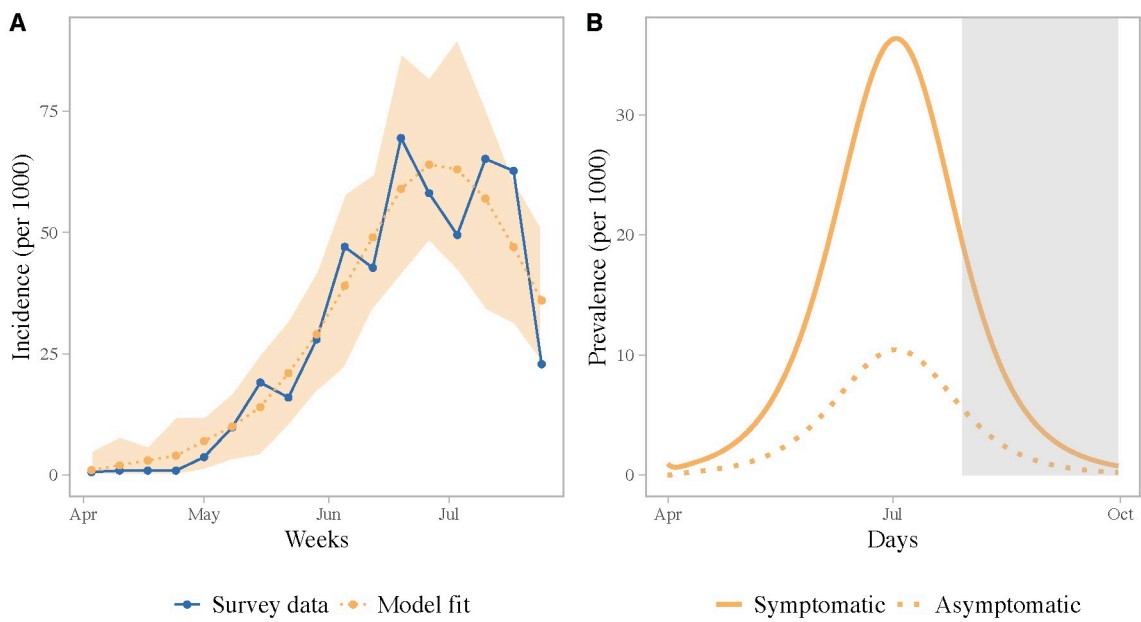

**Fig 2.** (A) Observed and fitted weekly incidence. The blue line shows the observed incidence from the survey. The orange dotted line shows the best-fit model. The shaded region shows the full range of incidence for 100 simulations assuming a poisson error structure. (B) Daily prevalence of symptomatic and asymptomatic infections simulated using the best-fit model parameters. The grey shaded region indicates the time period beyond the survey timeframe.

## Population mobility in Bangladesh and predicted importation from Dhaka

Fig 3 characterizes the population movement patterns in Dhaka, and travel from Dhaka to the rest of the country, derived from geo-located, anonymized and aggregated CDR data. We restricted the analysis to the area covered by the epidemiological survey, namely the core of Dhaka city. The mobile phone data for Dhaka represents about 4.4 million subscribers within the survey area, compared to a population size of 7.3 million. Within Dhaka, we found spatial heterogeneity in both mobility flux, a measure of average daily flow in and out of a location, and mobility network degree, a measure of the connectedness of each union. Specifically, neighborhoods in the northwest and southwest were more connected and had larger flows in and out of the area, proportional to their population size, compared to other locations. Increased connectivity, specifically having an additional degree in the mobility network, was associated with a 7% [95% CI: 0.5%, 14.5%] increase in the incidence rate of symptomatic cases (Fig 1C).

Large volumes of travel occurred daily between Dhaka and all other regions in the country, particularly on weekends and holidays. On average, 6% of subscribers within our study area traveled to other parts of the country daily. Fig 3D shows the daily proportion of subscribers leaving Dhaka. Weekends (Fridays and Saturdays) and the Eid holidays (which take place twice a year) can be clearly identified. During the start of the first Eid holidays, on June 23rd, approximately 1.7 million subscribers (40% of total subscribers) left Dhaka to visit other parts of the country.

We combined our mobility and prevalence estimates to predict the importation of chikungunya from Dhaka to the rest of the country. We compare our estimates to a standard diffusion model, to provide a comparison with predictions that would be made without mobile phone data, and where possible we compare our predictions to places that did report CHIKV

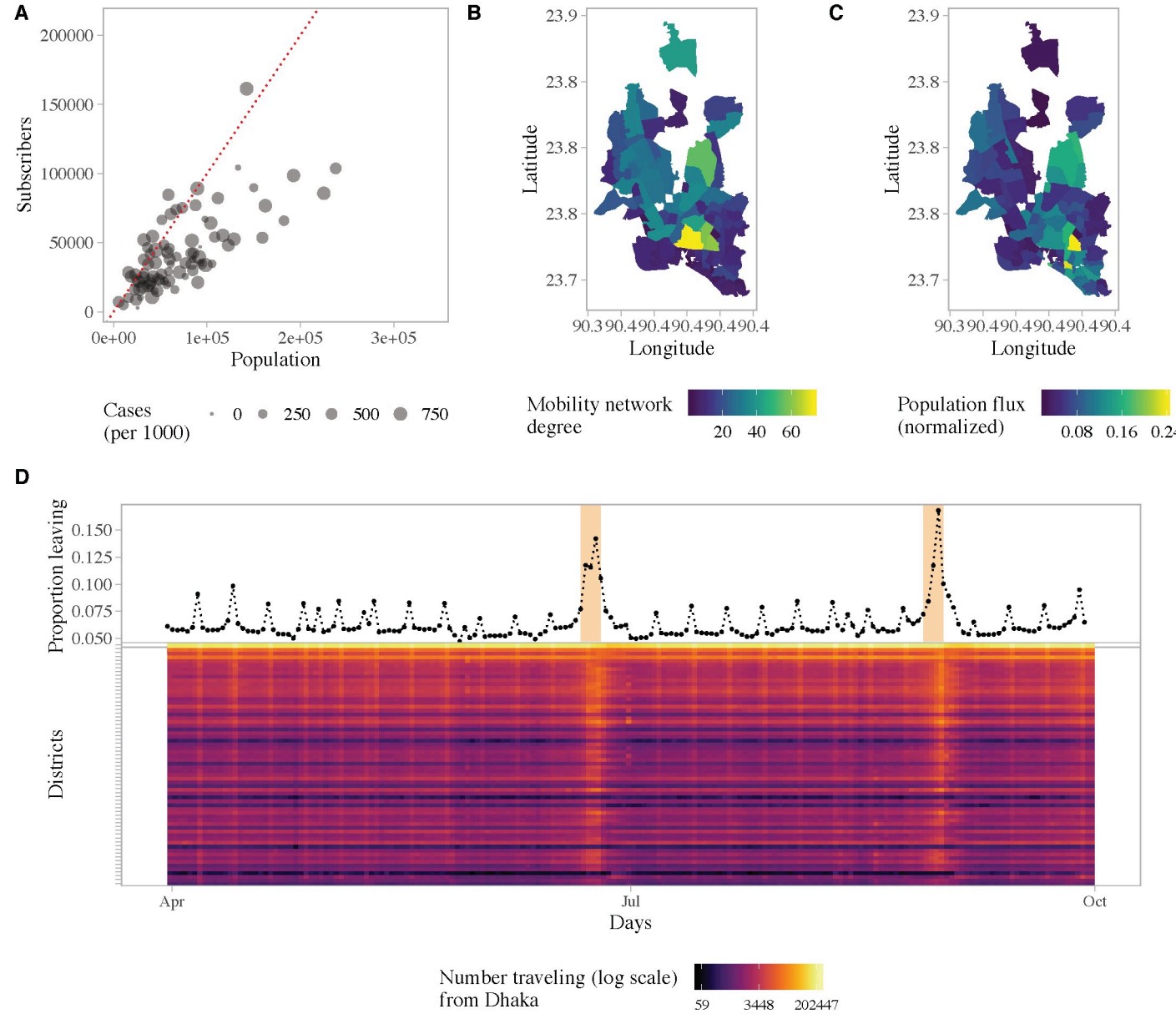

**Fig 3. Population movement patterns in Dhaka.** (A) Number of subscribers in a union versus the population size, for the unions covered by the survey. Map of Dhaka showing two measures of mobility: (B) the number of degrees in the network that had above average weight and (C) the average flow of people in and out of an area as a proportion of the population size. (D) Top panel: The daily proportion of subscribers who traveled from Dhaka to the rest of the country. This data represents the 4.4 million subscribers within the study area. Bottom panel: Daily number of subscribers (log count) traveling from Dhaka to all other districts. Districts are arranged from top to bottom by distance from Dhaka in ascending order.

cases during the outbreak, though these were limited. Our results highlight important deviations in actual travel patterns compared to predictions from a diffusion model. We predicted a high likelihood of importation into areas with large volumes of travel from Dhaka, particularly during the Eid holidays. Compared to the naive diffusion model, the model parameterized with mobile phone data showed much greater variation in imported cases across space and time (Figs 4 and 5). Specifically, for some districts that reported suspected cases, such as Bogra

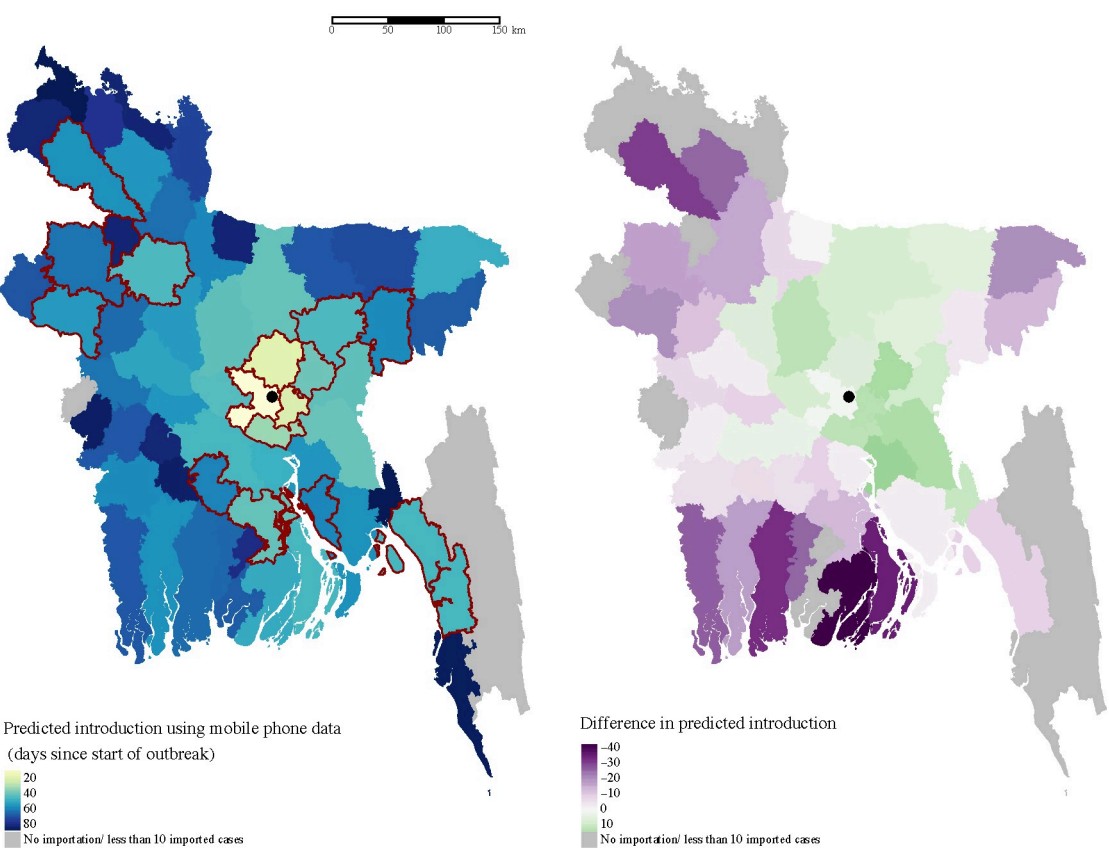

**Fig 4. Spatial heterogeneity in estimated importation time.** Introduction is defined here as the importation of at least ten cases, and measured as days since the start of the outbreak in Dhaka. The map on the left shows the estimated introduction time for each district based on mobility estimates. Lighter colors indicate earlier introduction. The districts that reported suspected cases to IEDCR are highlighted in red. The location of Dhaka city is indicated with the black circle. The difference between estimates using mobile phone data and a diffusion model are highlighted in the map on the right. In general, the diffusion model predicts earlier introduction (green) to nearby locations, and later (purple) or no introduction to more distant locations.

and Dinajpur, we predicted a larger number of imported cases than would be expected based on its distance from Dhaka. In the diffusion model importation decayed rapidly with distance, with very few imported cases to locations far from Dhaka. In contrast, the mobility estimates showed high volumes of travel to some locations, such as Dinajpur district, that are far from Dhaka. In general, the diffusion model predicted later introduction to places far from Dhaka, and earlier introduction to places closer to Dhaka, compared to the model with mobile phone data (Fig 4).

The Eid holidays were associated with a substantial increase in the estimated number of imported cases, which was not captured in the diffusion model (Fig 5). This reflects the large volume of travel from Dhaka to the rest of the country during the start of the Eid holidays. Our estimates of the number of imported cases is conservative, as we are only considering travel from the 95 unions included in the survey. In reality, the outbreak in Dhaka likely affected all unions in the greater metropolitan area, thus creating a much larger pool of potential infected travelers, especially during Eid.

Due to the lack of a surveillance and disease reporting system, only a handful of districts outside Dhaka reported cases to the Institute of Epidemiology, Disease Control and Research (IEDCR). Since reporting was not mandatory, there is no information about chikungunya

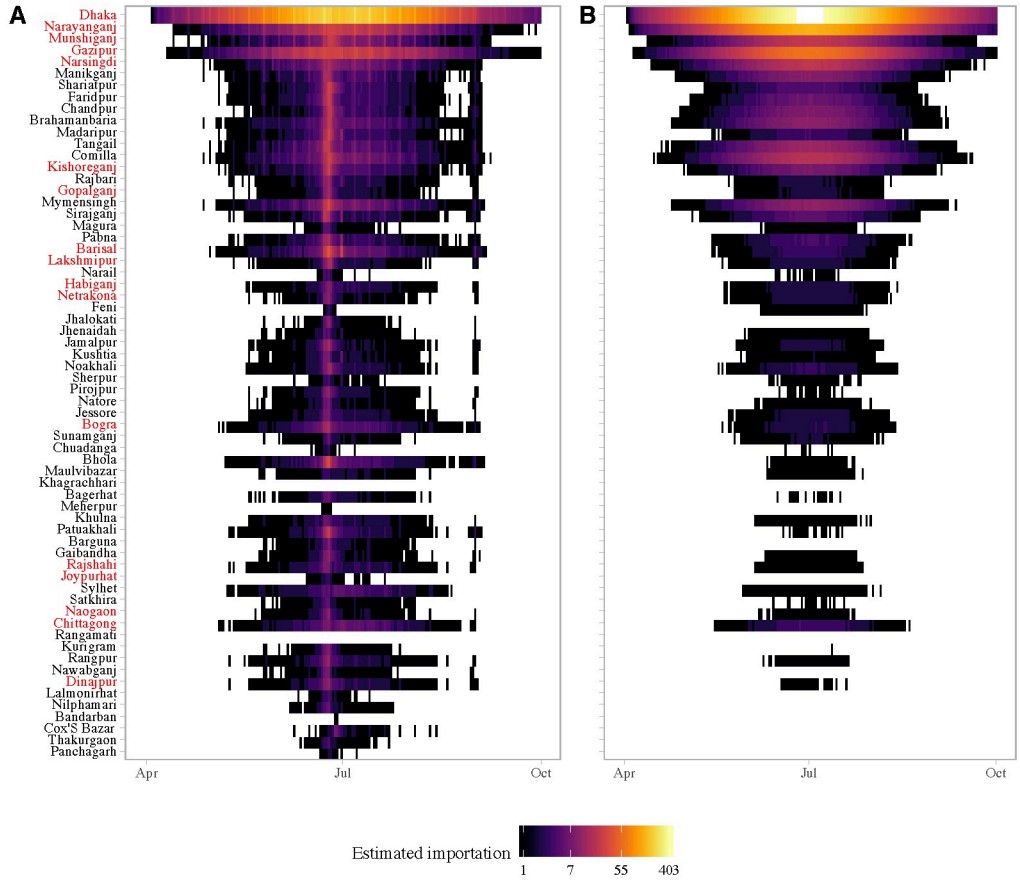

**Fig 5. Daily simulated imported cases (log count) from Dhaka to other districts in Bangladesh using (A) mobility estimates versus a (B) diffusion model.** Importations are simulated assuming only the asymptomatic infected are traveling ($p_{eff} = 0.1$). Each row represents a district; districts are arranged from top to bottom by distance from Dhaka in ascending order. The districts that reported suspected cases to IEDCR are highlighted in red.

incidence in the majority of districts in Bangladesh. However, the geographic distribution of the reporting districts (Figs 4 and S13) suggests that the outbreak had spread to many parts of the country, including districts far from Dhaka such as Dinajpur.

Eight of the 64 districts in Bangladesh, reported at least one probable chikungunya case, and 17 reported suspected cases, with the majority of cases reported in Narshingdi, a district close to Dhaka where we predicted a large number of imported cases from Dhaka. All the districts that reported five or more probable cases (Chittagong, Bogra, Dhaka, Munshiganj, Narshingdi) to IEDCR were amongst our top twenty districts based on the predicted cumulative number of importations from Dhaka. Our model also predicted importations to areas far from Dhaka, such as Joypurhat district, whereas the naive diffusion model predicted no importations. Cases from districts outside Dhaka were only reported for a short period of time, between July 17 and August 10, and thus, the start of the outbreak could not be inferred from the data to compare with our importation estimates. Nonetheless, the timing of cases is consistent with increased likelihood of importation during Eid, and the long serial interval ($\approx 23$ days [46]) for chikungunya.

## Discussion

The 2017 chikungunya outbreak in Dhaka illustrates the importance of population dynamics in shaping infectious disease transmission across spatial scales. Similar to chikungunya outbreaks in other settings [17, 47], our model results suggest that the outbreak in Dhaka was widespread, with a much higher attack rate than the official case counts suggest. Within the highly populated urban core of Dhaka, population density and mobility were important drivers of spatial heterogeneities in incidence. While the abundance of the vector was the most predictive factor, population dynamics, including density and connectivity, were positively associated with chikungunya incidence. Our mechanistic human-mosquito transmission model was able to accurately capture the dynamics of the infection in Dhaka. We estimated a human-to-human reproduction number of 4.20, and that about 22% of the infections were asymptomatic, which is similar to estimates in other settings [17, 35, 46, 47].

The 2017 chikungunya outbreak, which peaked during a major travel holiday, also serves as an illustrative case study for the impact of large-scale population movements on the spread of infectious diseases. While there is no data on incidence available from outside Dhaka, reports of suspected and probable cases from districts far from Dhaka, suggest that the outbreak had spread to other parts of the country by August. In contrast to the standard diffusion model, where importation rapidly decayed with distance from Dhaka, a model parameterized with mobility data predicted importations to most of the country. Detailed mobility analysis using the mobile phone CDR data revealed large fluctuations in population flows during the two major Eid holidays, during which millions of people traveled from Dhaka to other parts of the country. Such short-term population fluctuations are difficult to capture using traditional data sources and standard diffusion models, yet are likely to be crucial for the success of control efforts. Our results suggest that, given the large volumes of travel, outbreaks occurring during major holidays, such as Eid, are likely to spread rapidly. This is especially problematic for large megacities such as Dhaka, which are highly connected to the rest of the country and can amplify the spread of local outbreaks. Recent work suggests that the *Aedes* mosquito species exist in almost all parts of Bangladesh [48], thereby creating the opportunity for local outbreaks to be sparked by imported cases.

Our study has several limitations. First, our chikungunya incidence estimates are based on survey data rather than population-based surveillance. In the absence of routine surveillance, household surveys provide a cost-effective way to estimate incidence, but may not be representative of the whole population. Since symptoms were self-reported and not all suspected cases were lab-confirmed, our incidence estimates may be biased. However, the survey estimates of proportion seropositive were similar to independent estimates from other studies using a convenience sample [49] and hospital data [50], and the temporal pattern in cases reported to IEDCR was qualitatively similar to the estimated incidence from the survey [45]. Second, our estimates of population mobility may be biased due to differences in the sample of mobile phone users compared to the general population. While mobile phone owners may not be representative of the population, the large fraction of the city that the subscribers represent, suggests that it is at least capturing a significant component. The trend of massive increases in travel during the Eid holidays is unlikely to be an artifact of differential mobile phone ownership. Previous work in Kenya has also shown that mobility estimates from CDR data were not significantly affected by biases in ownership [51]. Comparing different data sources (e.g. from other operators or social media) on mobility in these populations will be an important next step for these approaches to be validated. Finally, our importation estimates cannot be validated against incidence data due to limited surveillance and reporting from outside Dhaka. Given the paucity of data on incidence outside Dhaka, we are also unable to parametrize a full

meta-population model; nonetheless, these results highlight important deviations of actual travel patterns from a standard diffusion model and the implications for the spatial spread of diseases in Bangladesh.

The 2017 chikungunya outbreak in Dhaka serves as a cautionary example of the potential for infectious diseases to spread rapidly from large urban centers, particularly if the timing of outbreaks coincide with major holidays. Combining novel sources of mobility data with epidemiological models, is a promising avenue for real-time forecasting during outbreaks, and would allow policy makers to target the highest risk populations. As populations become more mobile and connected, incorporating high resolution mobility data into models for forecasting will be crucial for identifying and targeting transmission hotspots, and for enacting control interventions. Our results suggest that seroprevalence studies should be prioritized in areas where importation of cases from Dhaka is highly likely. Forecasting the likelihood of importation into different areas could also provide an early-warning system for local health officials, particularly in anticipation of major travel events.

## Supporting information

**S1 Fig. Map of greater Dhaka city showing households surveyed by administrative unit.** The boundaries of the smallest administrative units (unions) are shown in white. Grey indicates locations that were not part of the survey.
(TIFF)

**S2 Fig.** (A) Weekly number of suspected (both tested and non tested) individuals, tested individuals, and confirmed cases in the survey. The ratio of confirmed to tested cases by (B) week and (C) location. While there may be true variation in this ratio across time and space, much of the sample variation appears to be driven by the small number tested in certain weeks or locations.
(TIFF)

**S3 Fig. Comparison of cases estimated through two methods.** X-axis shows the cases in each location estimated using the global sample ratio of confirmed cases to the number tested. Y-axis shows the cases estimated by drawing from a binomial distribution with size (number of trials) given by the number of untested suspected cases and the probability of a positive case given by the local ratio of confirmed cases to the number tested. The mean and standard deviation (error bars) across 100 random draws are shown here.
(TIFF)

**S4 Fig. Human-mosquito transmission model.**
(TIFF)

**S5 Fig. Observed and fitted weekly incidence.** The blue line shows the observed incidence from the survey. The orange dotted line shows the best-fit model. The model was fit to observed data assuming a measurement error structure i.e. we maximize the likelihood that the simulated incidence is drawn from a Poisson distribution with mean given by the observed data. The shaded region shows the full range of incidence for 100 simulations (using the best-fit parameters) and assuming a Poisson error structure.
(TIFF)

**S6 Fig. Parameter identifiability analysis.** Parameter estimates from fitting the model to simulated data. The simulated data was generated for ten sets of parameter values; *true* parameter values were randomly drawn from a uniform distribution with minimum and maximum values defined by the 95% CI for each parameter (from the main model fit). For each set of

parameter values, 100 incidence trajectories were simulated with a Poisson observational error structure. Median estimates from model fitting and standard deviation across the 100 simulations (error bars) are shown here. The red dashed line indicates the x = y line. In general, the standard deviation is smaller when the peak simulated incidence occurs earlier in the observation window i.e. more of the outbreak is observed.
(TIFF)

**S7 Fig. Sensitivity of model output to different values of each parameter.** Histogram of model output for (A) peak incidence and (C) the timing of the peak (in days from the start of the outbreak). Parameters were sampled via Latin Hypercube Sampling, and models were simulated using 100 parameter combinations. Partial rank correlation analysis for (B) peak incidence and (D) timing of the peak, to assess sensitivity of model output to each parameter. The height of the bar indicates the correlation of the model output with the given parameter, holding all else equal. Positive values indicate a positive change in the model output in response to an increase in the parameter, while negative values indicate a negative change. Parameters with greater influence have large absolute values of the correlation coefficient. *** indicates $p < 0.05$.
(TIFF)

**S8 Fig. The range of introduction times (shown as days since start of outbreak in Dhaka), for two extreme values of $p_{eff}$.** Introduction is defined here as the importation of at least ten cases. For each district, the lowest value (earliest introduction) represents $p_{eff} = 0.5$ and the highest value (latest introduction) represents $p_{eff} = 0.01$. The dots represent the $p_{eff} = 0.1$ scenario, which is used in the main results. For some districts, no importations were predicted with $p_{eff} = 0.01$; for these districts the highest value represents $p_{eff} = 0.1$.
(TIFF)

**S9 Fig. Daily simulated imported cases (log count) from Dhaka to other districts in Bangladesh, assuming only the asymptomatic infected are traveling ($p_{eff} = 0.01$).** Each row represents a district; districts are arranged from top to bottom by distance from Dhaka in ascending order.
(TIFF)

**S10 Fig. Daily simulated imported cases (log count) from Dhaka to other districts in Bangladesh, assuming only the asymptomatic infected are traveling ($p_{eff} = 0.5$).** Each row represents a district; districts are arranged from top to bottom by distance from Dhaka in ascending order.
(TIFF)

**S11 Fig. Spatial distribution of (A) the Breteau Index, defined as the number of containers that were positive for mosquito larvae per 100 houses inspected and (B) average population density (people per hectare).**
(TIFF)

**S12 Fig. Weekly precipitation (grey bars) and the estimated number of cases in the survey data (red).**
(TIFF)

**S13 Fig. Map of Bangladesh showing the number of suspected cases reported by districts across the country.** The red dot shows the location of Dhaka city. Cases in Dhaka district (highest number reported) are for locations in Dhaka district outside of the city limits.
(TIFF)

**S1 Table. Sensitivity of parameter estimates to the initial proportion susceptible.** The median and range of estimates for each of the fitted parameters are shown. The initial proportion susceptible was varied from 0.6 to 1 by increments of 0.05. The range of parameter estimates were obtained by fitting the simulated incidence from the mechanistic model to the observed incidence in the survey, assuming the different values for the initial proportion susceptible.
(PDF)

## Author Contributions

**Conceptualization:** Ayesha S. Mahmud, Md. Iqbal Kabir, Caroline O. Buckee.

**Data curation:** Ayesha S. Mahmud, Md. Iqbal Kabir, Kenth Engø-Monsen, Sania Tahmina.

**Formal analysis:** Ayesha S. Mahmud, Md. Iqbal Kabir.

**Investigation:** Ayesha S. Mahmud, Md. Iqbal Kabir.

**Methodology:** Ayesha S. Mahmud, Md. Iqbal Kabir, Caroline O. Buckee.

**Writing – original draft:** Ayesha S. Mahmud, Md. Iqbal Kabir, Kenth Engø-Monsen, Sania Tahmina, Baizid Khoorshid Riaz, Md. Akram Hossain, Fahmida Khanom, Md. Mujibor Rahman, Md. Khalilur Rahman, Mehruba Sharmin, Dewan Mashrur Hossain, Shakila Yasmin, Md. Mokhtar Ahmed, Mirza Afreen Fatima Lusha, Caroline O. Buckee.

**Writing – review & editing:** Ayesha S. Mahmud, Md. Iqbal Kabir, Kenth Engø-Monsen, Sania Tahmina, Baizid Khoorshid Riaz, Md. Akram Hossain, Fahmida Khanom, Md. Mujibor Rahman, Md. Khalilur Rahman, Mehruba Sharmin, Dewan Mashrur Hossain, Shakila Yasmin, Md. Mokhtar Ahmed, Mirza Afreen Fatima Lusha, Caroline O. Buckee.

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
