## [Decision Letter · Decision Letter 0]

21 Jul 2020

Dear Dr. Mahmud,

Thank you very much for submitting your manuscript "Megacities as drivers of national outbreaks: the 2017 chikungunya outbreak in Dhaka, Bangladesh" for consideration at PLOS Neglected Tropical Diseases. As with all papers reviewed by the journal, your manuscript was reviewed by members of the editorial board and by several independent reviewers. In light of the reviews (below this email), we would like to invite the resubmission of a significantly-revised version that takes into account the reviewers' comments. 

We cannot make any decision about publication until we have seen the revised manuscript and your response to the reviewers' comments. Your revised manuscript is also likely to be sent to reviewers for further evaluation.

Sincerely,

Christopher M. Barker

Associate Editor

Tereza Magalhaes

Deputy Editor

Reviewer's Responses to Questions

**Key Review Criteria Required for Acceptance?**

**Methods**

-Are the objectives of the study clearly articulated with a clear testable hypothesis stated?

-Is the study design appropriate to address the stated objectives?

-Is the population clearly described and appropriate for the hypothesis being tested?

-Is the sample size sufficient to ensure adequate power to address the hypothesis being tested?

-Were correct statistical analysis used to support conclusions?

-Are there concerns about ethical or regulatory requirements being met?

Reviewer #1: I think the objectives and hypotheses are clearly stated, and the methodology presented is relevant to the question. Nevertheless, as the authors claim that it is important to consider such mobility data in epidemiological model, I think it is required that the authors show what is the model best fit without considering these data. If the model can produce similar results without such mobility data, it would highlight that these data improve model fit, but is not a required component.

Reviewer #2: L70 : - the sample should be described more consistently: 3253 Hsld were included. How many individuals had a symptomatic history? 

1487 were sampled : what fraction of all cases is it? 1286 had IgM tested : what happened with the 201 other cases? 

Why were there some districts of Dhaka not included in the study?

L114 : the way you impute missing data reduces the heterogeneity in the percentage positive. The way O is computed 

may lead to values being overly influenced by the results of districts with large C and T. 

Using a mixed/random model with binomial sampling would be a better approach. This is all the more relevant since the association analysis shows 

may be anticonservative due to discounting uncertainty in reconstructing O. 

L113 : How long does it take for igM to be positive relative to infection & sampling? should we expect unerestimation here?

L179 : Definition of R0. Please indicate that this is the "human-specific" R0, i.e. human to human. 

The literature is not totally consistent regarding the definition of R0. 

However, the next generation method should yield the square root of the reported quantity, i.e human -> moquito (or mosquito -> human)

This is described in Roberts M.G., Heesterbeek J.A.P. A new method for estimating the effort required to control an infectious disease. Proceedings of the Royal Society of London B: Biological Sciences. 2003;270(1522):1359–1364 and in van der driessche papers.

L205: modelling incidence as a gaussian variable in the fit of the model is questionable. 

Poisson or NegBin is far more reasonable, as you did in the first part of the analysis, because variance is unlikely to be constant with incidence.

Does it make a difference if you fit the model using a Poisson ? 

Furthermore it seems that this is what you do in the computation of the confidence intervals. 

Is there a reason you use different models for analysing the data and performing parametric bootstrap?

In theory, parametric bootstrap is valid if your model is well specified. This seems unlikely 

 if it is not the same for simulation and estimation.

Another possibility would be to estimate parameter variance from Fisher's information matrix.

L209 : If you used the Nelder Mead algorithm, why did you require LHS sampling?

L211: I'm not convinced that both beta1 and beta2 can be estimated with the data that you have. 

You certainly have information on the product beta1 * beta2 that defines the growth 

rate of the epidemic curve as seen in the R0 formula. Can you provide simulation data showing that joint estimation is indeed possible with the data you have?

You further report that "The majority of the starting values converged to the same maximum-likelihood parameter estimate for 1 (median = 0.51, 213

25th percentile = 0.51, 75th percentile = 0.52), 2 (median = 0.16, 25th percentile = 0.16, 75th percentile = 0.16) and (median = 0.78, 25th percentile = 0.78, 75th 215 percentile = 0.78).".

Does this mean your likelihood present several maxima?

L213: regarding the rates, are the ranges OK? in normalized models with mosquito-borne transmission, 

we have beta1 m = beta2 where m is the number of mosquito per human. Furthermore, the betas correspond with 

frequency of bites, which is controlled by the gonotrophic cycle in mosquitoes. 

The mosquito to human rate is beta1 estimated at 0.51. Does this mean that an infected mosquito bites someone every 2 days? this is somewhat short given the gonotrophic cycle for Aedes, 

even in Dakha, should be more in the 3-4 days range. The human to mosquito means that an infected human is bitten every 1/0.16 days. With 4 days being infectious, 

this amounts to less than one bite on average during the infectious period. Some discussion of these numbers is necessary, if really they can be estimated.

**Results**

-Does the analysis presented match the analysis plan?

-Are the results clearly and completely presented?

-Are the figures (Tables, Images) of sufficient quality for clarity?

Reviewer #1: The results and the analyses are clearly presented

Reviewer #2: (No Response)

**Conclusions**

-Are the conclusions supported by the data presented?

-Are the limitations of analysis clearly described?

-Do the authors discuss how these data can be helpful to advance our understanding of the topic under study?

-Is public health relevance addressed?

Reviewer #1: I think the analysis only partially support their conclusions. As I've said before, it is important that the authors show how the model behave without these mobility data.

Reviewer #2: (No Response)

**Editorial and Data Presentation Modifications?**

Reviewer #1: I have no editorial modification to suggest

Reviewer #2: (No Response)

**Summary and General Comments**

Reviewer #1: Mahmud et al address the interesting question of how holiday travel can be involved in the spread of infectious diseases. Based on the outbreak of Chikungunya in Bangladesh in 2017, the authors have integrated mobile phone data within a mechanistic epidemiological model to estimate the contribution of population mobility into the spread of pathogens.

This is an interesting paper, but I need to be more convinced that data from mobile phone really improve model fit in this case. I'm a bit disapointed to see that only the dynamics in Dakha is really considered within the mechanistic model. It would have been much more relevant (I think) to consider a metapopulation model to model the whole spatio-temporal dynamics of this outbreak, moreover regarding the level of details of mobile phone data. This would have been a real novelty in the literature, and the kind of model that these data deserve.

Reviewer #2: - For the whole part on cell phone versus gravity model, little is shown to quantify that the mobility data 

actually does a better job to describe the data in bangladesh and to evidence that Dakha and mobility were 

the driver of the epidemic as indicated in the title.

Figure 5 shows the change in importation risk, but both quantities seems rather parallel. Is it possible to test that the 

places at risk are different and better predicted with the phone data?

You should try using the date and place of importation to come up with some evidence that cellphone data is more of interest than the gravity model, 

otherwise this remains hypothetical. Some more elaborated analysis is required for this part if you want to conclude on the use of cellphone data 

rather than another model of mobility in this particular instance.

- You report that up to 50% of the population was infected during the epidemic. However, you mention that 3 chikungunya epidemics 

already occured in the last 10 years. Given the attack rate that you estimated here, almost 50%, 

one would have expected that the percentage of the population that is actually susceptible would be far less than what you hypothesize. 

This should be discussed.

- all the data must be made available, in some useful aggregate form.

PLOS authors have the option to publish the peer review history of their article (what does this mean?). If published, this will include your full peer review and any attached files.

Reviewer #1: No

Reviewer #2: No
---

## [Decision Letter · Decision Letter 1]

4 Jan 2021

Dear Dr. Mahmud,

We are pleased to inform you that your manuscript 'Megacities as drivers of national outbreaks: the 2017 chikungunya outbreak in Dhaka, Bangladesh' has been provisionally accepted for publication in PLOS Neglected Tropical Diseases.

Best regards,

Christopher M. Barker

Associate Editor

Tereza Magalhaes

Deputy Editor

Reviewer's Responses to Questions

**Key Review Criteria Required for Acceptance?**

**Methods**

-Are the objectives of the study clearly articulated with a clear testable hypothesis stated?

-Is the study design appropriate to address the stated objectives?

-Is the population clearly described and appropriate for the hypothesis being tested?

-Is the sample size sufficient to ensure adequate power to address the hypothesis being tested?

-Were correct statistical analysis used to support conclusions?

-Are there concerns about ethical or regulatory requirements being met?

Reviewer #2: reply ok

Reviewer #3: All the objectives, design, sample size and statistical analysis are clearly stated and appropriate. There are none ethical concerns.

**Results**

-Does the analysis presented match the analysis plan?

-Are the results clearly and completely presented?

-Are the figures (Tables, Images) of sufficient quality for clarity?

Reviewer #2: reply ok

Reviewer #3: The results do match the study plan and design, they are clearly presented and all is sufficient and clear

**Conclusions**

-Are the conclusions supported by the data presented?

-Are the limitations of analysis clearly described?

-Do the authors discuss how these data can be helpful to advance our understanding of the topic under study?

-Is public health relevance addressed?

Reviewer #2: reply ok

Reviewer #3: Conclusions are supported, the limitations are clearly defined and the public health relevance is addressed

**Editorial and Data Presentation Modifications?**

Reviewer #2: reply ok

Reviewer #3: minor revision. Some sections like introduction needs to be review, since the text needs to be rearranged to allow for a better structure. For example, move the third paragraph up and summarize it a little more. In addition, some grammatical errors were detected in the manuscript, a full text revision is needed.

**Summary and General Comments**

Reviewer #2: reply ok

Reviewer #3: The study, to me, is solid and of public health relevance. No major revision is needed.

PLOS authors have the option to publish the peer review history of their article (what does this mean?). If published, this will include your full peer review and any attached files.

Reviewer #2: No

Reviewer #3: No

---

## [Editor Report · Acceptance letter]

28 Jan 2021

Dear Dr. Mahmud,

We are delighted to inform you that your manuscript, "Megacities as drivers of national outbreaks: the 2017 chikungunya outbreak in Dhaka, Bangladesh," has been formally accepted for publication in PLOS Neglected Tropical Diseases.

Best regards,

Shaden Kamhawi

co-Editor-in-Chief

Paul Brindley

co-Editor-in-Chief
